# Elasto-Plastic Fatigue Crack Growth Behavior of Extruded Mg Alloy with Deformation Anisotropy Due to Stress Ratio Fluctuation

**DOI:** 10.3390/ma15030755

**Published:** 2022-01-19

**Authors:** Kenichi Masuda, Sotomi Ishihara, Noriyasu Oguma, Minoru Ishiguro, Yoshinori Sakamoto

**Affiliations:** 1Department of Mechanical Engineering, University of Toyama, Gofuku 3190, Toyama 930-8555, Japan; sotomi.ishihara@gmail.com (S.I.); oguma@eng.u-toyama.ac.jp (N.O.); 2National Institute of Technology, Toyama College, Toyama 939-8630, Japan; mishiguro@nc-toyama.ac.jp (M.I.); ysa@nc-toyama.ac.jp (Y.S.)

**Keywords:** extruded Mg alloy, texture, elasto-plastic fatigue crack growth, stress ratio change, twin, fatigue crack closure, effective J-integral range

## Abstract

Fatigue crack growth (FCG) experiments were performed using a low-temperature extruded magnesium alloy AZ31 with texture. Under a constant maximum stress intensity factor (K_max_), the stress ratio R was changed from 0.1 to −1 during the fatigue crack growth process, and the FCG behavior before and after the R change was investigated. As a result, tensile twins were generated owing to the fatigue load on the compression side of R = −1, and the FCG velocity was accelerated. In addition, when the maximum compressive stress at R = −1 (|(σ_min_)_R = −1_|) exceeded the compressive yield strength of the material (σ_cy_), the FCG velocity after R fluctuation greatly accelerated. On the other hand, under the condition |(σ_min_)_R = −1_| < σ_cy_, the degree of acceleration of the FCG velocity due to R fluctuation was small. In either case, the degree of acceleration in the FCG increased as the K_max_ value increased. The above FCG acceleration mechanism due to the R fluctuation was considered based on the observation of the deformation and twinning states of the fatigue crack tip, the fatigue crack closure behavior, and the cyclic stress–strain curve of the fatigue process. The FCG acceleration mechanism was as follows: First, the driving force of the FCG increased owing to the increase in crack opening displacement due to the generation of tensile twins. Second, the coalescence of the main crack and a plurality of microcracks were generated at the twin interface. The elasto-plastic FCG behavior after the stress ratio fluctuations is defined by the effective J-integral range ΔJ_eff_.

## 1. Introduction

Magnesium (Mg) alloys are the lightest of all practical metals and have excellent properties such as specific strength, vibration absorption, and recyclability [1]. In addition, Mg is easily available because it is distributed worldwide in a variety of minerals. However, this alloy is inferior in cost and corrosion resistance [1].

In recent years, there has been an increasing momentum to use Mg alloys as structural materials to reduce the environmental load and save energy in equipment. The processing methods for Mg alloys include casting, die casting, extrusion, and rolling [2,3,4]. Recently, the use of wrought materials, such as extruded and rolled materials, is expected because of their excellent mechanical properties and few defects.

The crystal structure of Mg is a hexagonal close-packed lattice, and slip deformation at room temperature is limited to the basal plane of the crystal [1,2,3,4]. Therefore, in Mg alloys, slip deformation is unlikely to occur, and twin crystal deformation occurs instead [3,4,5]. In a rolled or extruded Mg alloy, the c-axis of the crystal is oriented in a direction perpendicular to the rolling or extrusion direction [6,7]. A so-called texture occurs. Therefore, the extruded Mg alloy is brittle and exhibits a behavior that is easily plastically deformed in a specific direction (plastic deformation anisotropy).

Many studies have been conducted on the fatigue behavior of Mg alloys. For example, the effect of grain refinement on improvements to the fatigue strength of extruded Mg alloys [8]; they reported that the fatigue strengthening mechanism via shot peening differs between Mg and 6Zn–0.5Zr alloy and Mg–10Gd–3Y–0.5Zr. In addition, changes in the residual stress caused by shot peening [9] were studied. It has been reported that shot peening significantly improves fatigue life. Furthermore, studies on the high-cycle fatigue characteristics of Mg AZ91 alloy under transverse loading ambiance have been conducted [10].

The effect of deformation anisotropy due to texture on low-cycle fatigue behavior was studied using rolled Mg alloy AZ31 [11]. Applying a fatigue load parallel to the rolling direction of a specimen caused it to yield easily during compression, whereas if the load was applied perpendicular to the rolling direction, the specimen yielded easily during tension. The effect of specimen orientation on the fatigue characteristics of rolled Mg alloys [12] has been studied. It was reported that the fatigue life of a specimen prepared in the direction perpendicular to the rolling direction was longer than that of the specimen prepared in the rolling direction.

In addition, the effect of the stress ratio R on the high cycle fatigue life of AZ61 extruded Mg alloy in the presence of deformation anisotropy was studied [13]. These studies reported that R containing a compressive load had a shorter fatigue life than that containing only a tensile load. In addition, the effect of fatigue anisotropy on the fatigue life of extruded and rolled Mg alloy materials was studied, and it was reported that the layered structure, which was formed parallel to the extrusion direction, had a significant effect on the fatigue life [14]. In addition, the effect of R on the fatigue crack growth (FCG) rate of Mg alloy AM60B [15] was studied. In this study, a modified model, based on the Walker model, was proposed.

Regarding the load history, the over-load and load sequence effects on the FCG properties of the extruded Mg alloy AZ31B were studied [16]. Compact tension specimens oriented in three different directions with respect to the extrusion direction were employed in this study. The influences of overloading and two-step high-low-sequence loading on FCG behavior were investigated in detail. For cast Al-Si-Mg alloys, the effect of the solidified structure after casting on the FCG properties was studied [17]. The microscopic mechanism for crack formation and FCG properties in high-pressure cast Mg alloys has been studied [18]. A study on the increase in intragranular FCG resistance due to supersaturated carbon in Fe-C alloys [19] was conducted using micro-notch technology.

Regarding the FCG behavior of Mg alloys, the FCG behavior and mechanism of rolled Mg alloy AZ31 [20] were studied. In addition, the effect of the layered microstructure on the FCG properties of extruded Mg alloys [21] was studied and it was reported that the FCG rate was reduced by the layered structure.

In addition, the effects of the amount of Mn and the texture [22] and load frequency [23] on the FCG characteristics of extruded Mg alloys have been studied so far.

As mentioned above, studies on the fatigue strength and FCG characteristics of Mg alloys with plastic deformation anisotropy have been reviewed. When Mg alloys are used as structural members, attention should be paid to the influence of the plastic deformation anisotropy caused by the texture on the fatigue strength and FCG behavior.

To date, most studies on the FCG properties of extruded Mg alloys have been conducted to the extent that linear elastic fracture mechanics (LEFM) can be applied. It was found that there are very few studies on the FCG behavior and fatigue crack closure behavior in the elasto-plastic region.

In this study, two types of FCG experiments were conducted using an extruded AZ31Mg alloy with texture. One is the FCG experiment under constant R and under the small-scale yielding condition (SSY), hereafter referred to as program [I]. The other is the FCG experiment under R fluctuation (0.1 to −1) under the condition of constant maximum stress intensity factor K_max_, hereafter referred to as program [II]. The FCG experiments in program [II] were further divided into two types. That is, condition (A) (|(σ_min_)_R = −1_| < σ_cy_) and condition (B) (|(σ_min_)_R = −1_| > σ_cy_). Here, |(σ_min_)_R = −1_| and σ_cy_ indicate the absolute values of the maximum compressive stress at R = −1 (after R fluctuation) and the compressive yield stress of the material, respectively.

From the FCG experimental results of program [II], under condition (A), the FCG velocity after R fluctuation accelerates, but the degree of acceleration is small. On the other hand, under condition (B), the FCG velocity after R fluctuation is greatly accelerated.

The acceleration mechanism of the FCG velocity after the R fluctuation was studied based on the twin crystal formation morphology at the fatigue crack tip, crack closure behavior, and cyclic stress–strain (σ–ε) curves. Furthermore, the FCG behavior in the constant R and R fluctuation experiments was analyzed based on the linear elastic and elasto-plastic fracture mechanics.

## 2. Material and Experimental Methods

### 2.1. Material

The test material was an extruded material of commercially available Mg alloy AZ31 (round bar with a diameter of 50 mm, AZ31Mg). The extrusion conditions were an extrusion temperature of 573 K, an extrusion ratio of 10, and an extrusion speed of 2 m/min. Table 1 presents the chemical composition of AZ31, as described in the mill sheet when the test material was purchased. As shown in the table, AZ31 contains approximately 3% Al and 1% Zn.

Figure 1 shows a microstructure photograph of the alloy (TD-ED surface). Here, ED is the extrusion direction. In the plane perpendicular to the extrusion direction, ND and TD are the specimen width direction and specimen thickness direction, respectively. The as-received material was a round bar; however, because a plate-shaped specimen was created, the amount of processing differed between the plate thickness direction and plate width direction. This can affect the microstructure of the material. Therefore, the TD and ND directions are displayed separately. As shown in the figure, the specimen material contained a mixture of fine and coarse crystal grains. The average crystal grain size was 9 μm, as measured using the linear cutting method. Owing to the high strength of the fine crystal grains and grain boundaries, the FCG velocity is reduced as the fatigue cracks pass through the above locations. Therefore, the FCG behavior and fatigue life greatly vary owing to the influence of the microstructure of the material. It is interesting to elucidate the mechanism by which the inhomogeneous microstructure shown in Figure 1 occurs and we would like to discuss this in future research. Figure 2 shows a pole figure for the basal plane {0002} ({0001}) before the fatigue test. The pole figure was measured using an X-ray diffractometer (XRD, Bruker AXS, D8 DISCOVER) and Schulz’s reflection method. XRD measurements were performed under a Cr-Kα ray, a tube voltage of 40 kV, a tube current of 30 mA, a divergence slit of 1.0°, and a light receiving slit of 4 mm, with an α scanning range of 15–90° and a β scanning range of 0–360°.

As shown in the figure, the bottom surface {0002} is oriented from the ND to the TD in the plane perpendicular to the ED direction. However, some bottom surfaces are oriented towards the ED direction. The cause of this is unknown; however, it could be due to the relatively low extrusion temperature of the test material, i.e., 573 K. Therefore, it can be seen that this test material has a strong texture owing to the extrusion processing. Other researchers [8,9] also reported that texture formation occurs when Mg alloy is extruded or rolled.

Figure 3a shows the stress (σ)–strain (ε) diagram during the static tension and compression tests of the AZ31Mg alloy. Measurements were performed twice under the same conditions. For σ and ε, engineering stress and strain were used, respectively. For convenience, the compressive stress and strain in the figure are displayed as positive values (dash line). A differential transformer-type displacement gauge attached to the universal testing machine was used for strain measurement. For the static tension and compression tests, round bar test pieces with a gauge radius of 5 mm and a gauge length of 5 mm, as shown in Figure 3b, were used. Screws were machined on the specimen grips at both ends. The load direction was in the ED direction. A columnar specimen with flat ends is usually used for the compression test but, for comparison, the same specimen shape as the tensile test was used in this study. Table 2 lists the mechanical properties of AZ31 obtained from the static experiments. As shown in Table 2 and Figure 3a, the compressive yield stress σ_cy_ was 130 MPa, which is approximately 40% lower than the tensile yield stress σ_ty_ (210 MPa). There was a large difference in the yield strength between tensile and compression, showing deformation anisotropy [24,25,26,27,28].

### 2.2. Experimental Methods

#### 2.2.1. Program [I]: FCG Experiment under the Constant R Condition (Penetration Crack, SSY Condition)

A single-edge notched tensile (SENT) plate specimen (thickness of 4 mm) with a notch on one side, having the shape and dimensions as shown in Figure 4, was prepared from an extruded round bar with a diameter of 50 mm. The SENT specimens had a sharp notch with a length of 4 mm on one side; and were machined so that the load direction (specimen axial direction) was in the ED direction. To facilitate the observation of fatigue cracks, the specimen surface was mirror-finished with abrasive paper (SiC paper, # 800–2000) and diamond paste, and then subjected to a fatigue experiment.

To investigate the FCG characteristics of the extruded AZ31Mg alloy material, a fatigue experiment was conducted at room temperature (20–30 °C) and in air under cyclic stress speeds of 5–20 Hz and a sinusoidal load waveform. Using an electric/hydraulic servo-type fatigue testing machine with a capacity of 20 kN, tension–compression fatigue tests were performed. FCG experiments in program [I] were performed under constant R conditions (R = 0.1, R = −1). In addition, the FCG experiment was carried out under the condition that the cyclic plastic zone size of the crack tip was adequately smaller than the crack length, that is, the SSY condition.

The FCG experiments were conducted under increasing and decreasing conditions of ΔK. Most FCG experiments were performed under constant load conditions (ΔK-increasing test). In the low ΔK region, the ΔK-decreasing test was performed in some experiments but the decrease in ΔK was maintained within 2–3%. To calculate K of the SENT specimen, Equation (1) [29] was used.
(1)$$K=\sigma \sqrt{\pi a}\cdot \left(1.12-0.231\alpha +10.55\alpha^{2}-21.72\alpha^{3}+30.39\alpha^{4}\right),\;\alpha =a/W$$

Here, *a* and *W* represent the crack length and specimen width, respectively. The replica method [30] was used to measure *a*. The FCG experiment was interrupted after a certain number of stress cycles, and replicas of the specimen surface during the fatigue process were taken using an acetyl cellulose film and methyl acetate. Then, using an optical microscope, the crack length, a, recorded on the replica, was measured at a magnification of 100–200 times.

As can be seen from Equation (1), under the constant σ condition, the K value increases as the crack length *a* increases. Therefore, the constant ΔK experiment was performed by adjusting σ according to the elongation of *a*. The crack opening load P_op_ was measured by attaching a strain gauge in front of the crack tip (approximately 2 mm) and using the elastic compliance and subtraction methods [31]. The crack opening stress σ_op_ was obtained from P_op_, and the stress intensity factor K_op_ at the time of crack opening was determined by substituting σ_op_ into Equation (1). The obtained K_op_ was substituted into ΔK_eff_ = K_max_−K_op_ to determine the effective stress intensity factor ΔK_eff_. 

#### 2.2.2. Program [II]: FCG Experiment under R Fluctuation

By applying a fatigue load, a fatigue pre-crack was generated from the notch bottom of the SENT specimen, as shown in Figure 4. Then, the notch length (4 mm) and the opposite side of the notch (length 4 mm) of the SENT specimen were removed using an electric discharge machine. However, the fatigue pre-crack (approximately 0.1–0.3 mm) was not deleted and remained. The reason for the deletion is to adjust the crack length to be short, to facilitate the FCG experiment in the elasto-plastic state, and to facilitate the control of the K value. Therefore, in the FCG experiment under R fluctuation, the SENT specimen with a width of 17 mm, gauge length of 22 mm, and thickness of 4 mm was used.

In the FCG experiment under the R fluctuation test of program [II], R was changed from 0.1 to −1 under the constant K_max_ condition; at that time, the changes in the FCG behavior and morphology of the specimen surface due to the R fluctuation were investigated.

FCG experiments under R fluctuation were performed under two conditions: (A) (|(σ_min_)_R = −1_| < σ_cy_) and (B) (|(σ_min_)_R = −1_| > σ_cy_). Changes in the specimen surface due to R fluctuations were observed using a scanning electron microscope (SEM, Hitachi Co Ltd. (Tokyo, Japan) TM-1000).

Furthermore, an approximate evaluation of the J-integral was performed using the cyclic σ-ε curve.

(a) Cyclic σ-ε curve

In the FCG experiment under R fluctuation, the cyclic σ-ε curve was measured in addition to the FCG behavior. The measurement was performed by capturing electrical signals from a load cell (σ) and a strain gauge (ε) into a computer at each stress cycle over a constant number of stress cycles. A strain gauge was attached to the center of the specimen in the load direction. A data collection system (manufactured by KEYENCE) was used for the signal measurements.

(b) Approximate evaluation of J-integral

In program [II], some FCG experiments deviate from the SSY condition, and the linear elastic fracture mechanics (LEFM) parameter K cannot be used. Therefore, the elasto-plastic fracture mechanics (EPFM) parameter, J-integral, was used.

Equation (2) was used to evaluate the J-integral. The first and second terms on the right-hand side of Equation (2) represent the elastic and plastic components of the J-integral, respectively. The plastic component is represented by a simple evaluation formula, as obtained by Rice et al. [32] for the middle tension panel (hereinafter referred to as the MT specimen). Hoshide et al. [33] reported that this simple formula can be applied to short cracks.

The simple evaluation formula of the J-integral for the SENT specimen is unknown so, instead, the formula for the MT specimen is used in this study.
(2)$$J=\frac{K^{2}\left(1-\nu^{2}\right)}{E}+\left[\int_{0}^{u}\mathrm{Pdu}-\frac{1}{2}\mathrm{Pu}\right]/\left\{B\left(W-a\right)\right\}$$

Here, ν is the Poisson’s ratio of the material. The P and u terms are the load and opening displacements at the center of the crack, respectively. In the case of the MT specimen, B is the thickness of the specimen, W is the half-width of the specimen, and *a* is the half-length of the crack.

## 3. Experimental Results

### 3.1. Program [I]: FCG Experiment under the Constant R Condition

Figure 5 shows the relationship (blue data) between the FCG velocity da/dN and stress intensity factor range ΔK of the AZ31B extruded material (extrusion temperature of 573 K). Here, N is the number of stress cycles, and ΔK is given by (K_max_—K_min_), where K_min_ is the minimum stress intensity factor.

In the figure, the experimental results for R = 0.1 and −1 are plotted. For R = −1, we set ΔK = K_max_, assuming that the negative stress fluctuation range does not contribute to the crack growth.

The red experimental data (● and ○) in the figure are the FCG experimental results of the 693 K extruded material of AZ31B reported in a previous study [34] and are given for comparison with the experimental results of the 573 K extruded material.

The 693 K extruded material is the same material as the 573 K extruded material, and only the extrusion temperature is different. Its average crystal grain size is 32 μm, which is greater than the 9 μm grain size of the 573 K extruded material. The compressive and tensile yield stresses of the 693 K extruded material were 85 and 180 MPa, respectively [34]. These values are smaller than those of the 573 K extruded material in Table 2. It is presumed that this is mainly due to the difference in the crystal grain size between the two. The solid and dash lines in the figure show the FCG results for AZ31B rolled material (R = 0.05) [20], and AZ31B extruded material (R = 0.1) [35], respectively.

As can be seen from the figure, da/dN at R = −1 is accelerated compared to that at R = 0.1. The degree of acceleration was large in the low-ΔK region. The threshold value of the FCG velocity ΔK_th_ (da/dN ≅ 10^−9^ m/cycle) at R = −1 is considerably lower than that at R = 0.1. In addition, the da/dN−ΔK relationships are slightly different owing to the influence of the extrusion temperature (difference in crystal grain size). However, in the low-ΔK region, the characteristic that da/dN for R = −1 is faster than that for R = 0.1 is approximately the same. The experimental results of Tokaji et al. [20] and Morita et al. [35] are in good agreement with the da/dN−ΔK relationship at R = 0.1 in this study.

Figure 6 shows the relationship between da/dN and the effective stress intensity factor range ΔK_eff_ (= K_max_−K_op_) of this test material. For comparison purposes, the previously reported da/dN–ΔKeff relationship of the 693 K extruded material [34], and the same relationship of Tokaji et al. [20] with respect to AZ31B rolled material are also shown. As can be seen from the figure, the influence of R is not recognized in the da/dN–ΔK_eff_ relationship, regardless of the extrusion temperature. In addition, except for the low-ΔK_eff_ region, the influence of the extrusion temperature was not observed in the da/dN–ΔK_eff_ relationship. The threshold value of FCG, ΔK_effth_, of this test material is 0.6 to 0.7 MPam^1/2^. In addition, the da/dN–ΔK_eff_ relationship in this experiment is approximately the same as that of the AZ31B rolled material obtained by Tokaji et al. [20].

### 3.2. Program [II]: FCG Experiment under R Fluctuation (0.1 to −1)

#### 3.2.1. FCG Behavior

An FCG experiment under a constant ΔK was conducted, and the effect of R fluctuation on FCG behavior when R was changed from 0.1 to −1 was investigated during the experiment. The K_max_ values before and after the fluctuation in R were set to be equal.

Figure 7A shows the results of the R fluctuation experiment under condition (A) (|(σ_min_)_R = −1_ | < σ_cy_). Figure 7A(a–c) show the results for K_max_ values of 3, 4, and 5 MPa∙m^1/2^, respectively. However, Figure 7B shows the results of the R fluctuation experiment under condition (B) (|(σ_min_)_R = −1_ | > σ_cy_). Figure 7B(d–f) show the results for K_max_ values of 3, 4, and 5 MPa∙m^1/2^, respectively.

The vertical dashed lines in these figures indicate the time points of change in R (0.1 to −1). Because the FCG experiment is conducted under a constant K_max_, the *a*–N relationship can be approximated by a straight line, and its slope represents da/dN.

The FCG velocities, da/dN, before and after the R fluctuation, were obtained from Figure 7A,B. The results are summarized in Table 3. In the table, (da/dN)_R = 0.1_, and (da/dN)_R = −1_ are da/dN at R = 0.1 and R = −1, respectively, and (da/dN)_R = −1_/(da/dN)_R = 0.1_ indicates the acceleration rate of da/dN due to R fluctuation.

Figure 8 shows the effect of K_max_ on (da/dN)_R = −1_/(da/dN)_R = 0.1_ using the numerical values in Table 3. As can be seen from Figure 8 and Table 3, under condition (A), the acceleration rate was approximately 1–1.31 times, and for condition (B), the acceleration rate was approximately 2.5–4.0 times. Under either condition, the FCG velocity is accelerated by the fluctuation of R from 0.1 to −1. However, the acceleration rate of the latter was clearly higher than that of the former. In addition, under any condition, the larger the K_max_ value, the higher is the acceleration rate.

Next, K_op_ was measured using the elastic compliance method [28], and the ΔK_eff_ was calculated. Figure 9 shows the relationship between da/dN and ΔK_eff_ before and after R fluctuation. In the figure, the FCG data (○) under constant R tests are plotted for comparison.

Under condition (A) in Figure 9a, the results (●, ▲, ■) at R = 0.1 before the R fluctuation are consistent with the experimental data (○) under the constant R condition.

At R = −1 after R fluctuation, the experimental data (○, △) for K_max_ of 3 and 4 MPam^1/2^ agree with the da/dN–ΔK_eff_ relationship (○) under the constant R condition. However, at K_max_ = 5 MPam^1/2^ (□), as indicated by the elliptical mark, they are above the da/dN–ΔK_eff_ relationship (○) under constant R conditions, therefore indicating acceleration.

On the other hand, under condition (B) in Figure 9b, the experimental data (●, ▲) before the R fluctuation (R = 0.1) are approximately the same as the da/dN–ΔK_eff_ relationship (○) under the constant R condition. However, even before the R fluctuation (R = 0.1), the data (■) at K_max_ = 5 MPam^1/2^ exceeded the da/dN–ΔK_eff_ relationship (○), which was obtained under the constant R condition (SSY condition). The FCG velocity accelerated, as indicated by the oval mark in the figure. The reason for this is that the crack length is short and the load is large, resulting in a deviation from the SSY condition.

The data (○, △, □) at R = −1 after R fluctuation is located above the da/dN–ΔK_eff_ relationship (○) under the constant R condition, as shown by the elliptical marks in the figure. The reason for the acceleration is presumed to be the occurrence of tensile twins, which will be discussed in detail in Section 4.1. As mentioned above, the acceleration of the FCG velocity was remarkably observed at R = −1 after the R fluctuation.

#### 3.2.2. Stress–Strain Behavior (σ-ε Curve) under Repeated Stress

Figure 10 shows the cyclic σ-ε curve when R fluctuates (from 0.1 to −1) in the SENT specimen. The two loops labeled ‘R = 0.1’ and ‘R = −1’ are the σ-ε curves corresponding to immediately before (R = 0.1) and after (R = −1) of the R fluctuation in Figure 7. The σ-ε curves were used to calculate S that will be described in Section 4.2. The horizontal dash line in the figure indicates the compressive yield stress (σ_cy_ = 130 MPa) of the test material.

Figure 10a,b correspond to K_max_ values of 4 MPa∙m^1/2^ and 5 MPa∙m^1/2^, respectively. These results are the experimental results for condition (A). As can be seen from these figures, under condition (A), at R = −1 after the R fluctuation, the cyclic σ-ε curve (red) shows a hysteresis loop. However, its degree is small compared to that of R = 0.1 (black) without a hysteresis loop. Therefore, the SSY condition is almost satisfied. The above experimental tendency indicates that the da/dN−△K_eff_ relationship after the R fluctuation is consistent with that under the constant R condition (Figure 10a, SSY condition).

Figure 10c,d correspond to K_max_ values of 4 MPa∙m^1/2^ and 5 MPa∙m^1/2^, respectively. These are the experimental results for condition (B), that is, the large-scale yielding condition.

As shown in the figures, under condition (B), clear hysteresis loops were observed in the cyclic σ-ε curves (red) at R = −1 after the R fluctuation. Furthermore, the hysteresis loop at R = −1 has an asymmetric shape on the tension and compression sides, and the hysteresis area A_C_ on the compression side is greater than A_T_ on the tension side.

The hysteresis loop area ratio A_T_/A_C_ was 0.59 at K_max_ = 4 MPa∙m^1/2^ (Figure 10c) and 0.53 at K_max_ = 5 MPa∙m^1/2^ (Figure 10d). A_T_/A_C_ = 1 indicates that the areas of the tension and compression sides of the hysteresis loop are equal. Therefore, at K_max_ = 5 MPa∙m^1/2^, the asymmetry of the hysteresis area appears more strongly than at K_max_ = 4 MPa∙m^1/2^.

The experimental tendency for the hysteresis loop in condition (B) corresponds to the fact that the da/dN−ΔK_eff_ relationship after the R fluctuation was accelerated more than that under the constant R condition (SSY condition) (Figure 9b).

The slope of the curve at the time of unloading can be linearly approximated. However, its slope is somewhat less than that of Young’s modulus (E = 45 GPa) in the static strength experiment, probably due to cyclic softening, and is approximately 38 GPa.

The hysteresis loop and its asymmetry in the cyclic σ-ε curves at R = −1 after the R fluctuation are presumed to be due to the tensile twins generated at the time of compressive loading, and the details are discussed in Section 4.1.

#### 3.2.3. Changes in the Appearance of the Crack Tip Region Due to R Fluctuation

Figure 11 shows the appearance of the fatigue crack tip during the R fluctuation observed using a scanning electron microscope (SEM). Observations were performed under condition (B), K_max_ = 5 MPa∙m^1/2^ (Figure 10d). The fatigue crack grew from left to right. The white arrow in the figure indicates the R fluctuation point, R = 0.1–1.

As can be seen from the figure, at R = −1 after the R fluctuation, a large deformation occurs in front of the crack tip, and multiple microcracks occur in the area surrounded by the dotted line, as indicated by the arrow. Such a morphology is clearly different from the morphology before the R fluctuation (R = 0.1) in the same figure. On the other hand, although the figure is omitted, under condition (A), even if the same K_max_ = 5 MPa∙m^1/2^, the change in the crack tip due to the R fluctuation was not clear.

In addition, although the figure is omitted, the same R fluctuation experiment was performed on the surface cracks generated on the smooth specimen without notches (same material). As a result, a large deformation and microcrack formation were observed at the crack tip after the R fluctuation, and a similar aspect to the above-mentioned penetrating crack was observed.

The above-mentioned changes in the appearance of the crack tip after the R fluctuation show a good correspondence with the FCG behavior (Figure 7 and Figure 9) and the behavior of the hysteresis loop (Figure 10). The details are discussed in Section 4.1.

## 4. Discussions

### 4.1. On the Acceleration Mechanism of da/dN and the Hysteresis Loop in the σ-ε Curve, When R Fluctuates

#### 4.1.1. Acceleration Mechanism of da/dN

After R fluctuation, da/dN at R = −1 accelerated compared with that at R = 0.1. It is presumed that this acceleration in da/dN occurred through steps ①–④, as shown in Figure 12. The c-axis before the fatigue load faces in the direction of ND-TD in the plane orthogonal to the fatigue load direction, as shown in Figure 2.

➀: At R = −1, when a compressive load acts in the ED direction, tensile deformation occurs on the c-axis, which is oriented in the ND-TD direction.

②: When tensile deformation occurs in the c-axis direction, {101 ®2} < 101 ®1 > tensile twins are generated. Owing to the generation of tensile twins, the c-axis changes direction by approximately 90° from the direction before fatigue loading (ND-TD direction) to the ED direction [34].

③: When the c-axis is oriented at 86.3°in the ED direction owing to the generation of tensile twins, shear stress acts on the basal plane {0002} in addition to normal stress due to fatigue load, leading to the basal plane dislocation operation. As a result, fatigue cracks develop along the basal plane {0002}.

④: The generation of tensile twins promotes the deformation of the specimen and the generation of microscopic cracks (Figure 11). The crack opening displacement (COD) is given by COD ∝ K^2^/(Eσ_cy_), where E is Young’s modulus. When a compression load is applied at R = −1, the yield strength σ_cy_ decreases owing to twinning. In addition, it is expected that E will decrease owing to the cyclic softening behavior during the fatigue process. As a result, the COD behind the crack tip increased. Increasing the COD reduces K_op_ (J_op_ in the elasto-plastic region), resulting in an increase in ΔK_eff_ (ΔJ_eff_ in the elasto-plastic region). Here, J_op_ and ΔJ_eff_ are the J-integral when the crack opens and the effective J-integral range, respectively. As a result, da/dN is considered to be accelerated. These are described in Section 4.2.

Another possible factor for the acceleration of da/dN is the coalescence of microcracks and the main crack. As shown in Figure 11, numerous microcracks occurred at the twins. As shown in the schematic diagram of Figure 12, it is considered that the main cracks coalesced with these small cracks, leading to acceleration of the FCG velocity.

#### 4.1.2. Behavior of Hysteresis Loop on the Cyclic σ-ε Curve

Under condition (B), when R changes from 0.1 to −1, a large number of twins are generated in the specimen, and the area of the hysteresis loop in the cyclic σ-ε diagram increases (Figure 10c,d).

Furthermore, as can be seen from the observation result (Figure 11), a large deformation occurs at the crack tip after the R fluctuation, and the SSY condition is not satisfied. Therefore, in the R fluctuation of condition B, the FCG velocity accelerates (Figure 7B(d–f) because of the large deformation of the crack tip.

Moreover, because the SSY condition is not satisfied, the FCG velocity after the R fluctuation cannot be described by ΔK_eff_, as shown in Figure 9b. On the other hand, under condition A, when R changed, twins were limitedly generated near the tip of the crack. Therefore, the change in FCG velocity after R fluctuation is small (Figure 7A(a–c), and the area of the hysteresis loop is small (Figure 10A(a,b). The deformation near the crack tip approximately satisfies the SSY condition, and the FCG velocity after the R fluctuation can be described by ΔK_eff_ (Figure 9a).

### 4.2. Analysis of Elasto-Plastic FCG Behavior by J-Integral

Under condition (B), and condition (A) at K_max_ = 5 MPa∙m^1/2^, the vicinity of the crack tip was considered to be in a large-scale yielding condition. Therefore, the FCG behavior after R fluctuation is analyzed using the effective J-integral range ΔJ_eff_, which is an elasto-plastic fracture mechanics parameter.

Because the simple evaluation formula of the J-integral for the SENT specimen is unknown, in this study, the simple evaluation formula of the J-integral for the MT specimen —obtained by Rice et al. [32]—was used as an alternative formula. The degree of approximation of the MT and SENT specimens was examined, using K instead of J.

The evaluation of K for the MT specimen is given by the following equation [29].
(3)$$K\;=\;\sigma \sqrt{\pi a}\sqrt{sec\frac{\pi a}{2W}}$$

After setting the value of σ to an arbitrary value (for example, 100 MPa), for multiple values of a/W, the K values for the MT specimen (Equation (3)) and those of the SENT specimen (Equation (1)) were compared.

The K value of the MT specimen was approximately 10% lower than that of the SENT specimen and the difference between the two was not large. This result seems reasonable because both specimens are geometrically symmetric.

If we assume that the difference between the MT specimen and the SENT specimen in the simple evaluation formula of the plastic component of the J integral is about the same as the difference in K, it is considered that Equation (2) can also be used for the SENT specimen.

In this study, the opening displacement *u* at the center of the crack was approximated as the displacement at the load point [33]. In addition, in the SENT specimen in this study, *W* and *a* represent the specimen width and crack length, respectively. When the above approximations are made, Equations (4) and (5) are established between *P*, *u*, σ, and ε.
(4)$$\sigma =\frac{P}{\mathrm{BW}}\approx \frac{P}{B\left(W-a\right)}$$
(5)$$\;u=\frac{l}{2}\varepsilon$$

Here, ℓ is the length (22 mm) of the parallel portion of the specimen. In the SENT specimen shown in Figure 4, the left and right parts of the notch are symmetrical; therefore, in Equation (5), half of the length of the parallel part of the specimen was used. It was then approximated that only the parallel portion of the specimen with the minimum cross-sectional area was deformed under load, and the displacement *u* of the loading point was calculated. Substituting Equations (4) and (5) into Equation (2) gives the following equation for J:(6)$$J=\frac{K^{2}\left(1-\nu^{2}\right)}{E}+\frac{l}{2}\left[\int_{0}^{\varepsilon }\sigma d\varepsilon -\frac{1}{2}\sigma \varepsilon \right]$$

When Equation (6) is expressed as ΔJ_eff_ ( = J_max_ − J_op_), Equation (7) is obtained. Here, J_max_ and J_op_ indicate the J values at the maximum and at the time of crack opening, respectively.
(7)$$\Delta J_{\mathrm{eff}}=\frac{\Delta K_{\mathrm{eff}}{}^{2}\left(1-\nu^{2}\right)}{E}+\frac{l}{2}\left[\int_{\varepsilon_{\mathrm{op}}}^{\varepsilon_{\max}}\sigma d\varepsilon -{\left[\frac{1}{2}\sigma \varepsilon \right]}_{\sigma_{\mathrm{op}}}^{\sigma_{\max}}\right],\;\;\sigma_{\mathrm{op}}\leq \sigma \leq \sigma_{\max}$$

In Equation (7), σ_op_ and ε_op_ indicate the stress and strain at the time of crack opening, respectively, and σ_max_ and ε_max_ indicate the maximum stress and maximum strain, respectively. The first and second terms on the right-hand side of Equation (7) are the elastic and plastic components of ΔJ_eff_, respectively. The latter plastic component was determined using the cyclic σ-ε diagram shown in Figure 10.

The outline of the measurement is schematically shown in Figure 13. The area S (Equation (8)) of the region shown in gray in the figure was measured and multiplied by ℓ/2 to obtain the plastic component of ΔJ_eff_. S is obtained by subtracting the area of the triangle indicated by the arrow in the figure from the value obtained by integrating the cyclic σ-ε curve from ε_op_ to ε_max_. However, it should be noted that the measurement region is in the range of σ_op_ ≤ σ ≤ σ_max_. Area measurement software was used to measure S.
(8)$$S=\left[\int_{\varepsilon_{\mathrm{op}}}^{\varepsilon_{\max}}\sigma d\varepsilon -{\left[\frac{1}{2}\sigma \varepsilon \right]}_{\sigma_{\mathrm{op}}}^{\sigma_{\max}}\right],{\;\sigma }_{\mathrm{op}}\leq \sigma \leq \sigma_{\max}$$

Figure 14 shows the da/dN−ΔJ_eff_ relationship under R fluctuation. The plastic component of ΔJ_eff_ was obtained by measuring the area S in Figure 13. The hysteresis loops for K_max_ = 4 MPam^1/2^ and K_max_ = 5 MPam^1/2^ in Figure 10 were used as the experimental data.

The data points of ▲, ■, ▲, and ■ in the figure are the da/dN−ΔJ_eff_ relationship at R = 0.1 before the R fluctuation. The data points of △, □, △, and □ indicate the da/dN−ΔJ_eff_ relationship with respect to R = −1 after the fluctuation of R. In the figure, the converted results of the da/dN–ΔK_eff_ relationship (Figure 6) obtained from the FCG experiment under constant R conditions into the da/dN–ΔJ_eff_ relationship are indicated by the data points denoted by ○. For the conversion from ΔK_eff_ to ΔJ_eff_, the first term on the right-hand side of Equation (6) was used. The solid line is the result of converting the da/dN–ΔK_eff_ relationship (SSY condition is satisfied) obtained by Tokaji et al. to the da/dN–ΔJ_eff_ relationship using the same method as above. It was confirmed that the above conversion method from ΔK to ΔJ is appropriate by Endo et al. [36].

As shown in Figure 14, the da/dN−ΔJ_eff_ relationships before and after the R change agree with those under constant R conditions (○) and Tokaji et al. (solid line) [20].

In a previous study [34], using an extruded Mg alloy with a texture, the FCG behavior under SSY conditions was analyzed. As a result, under SSY conditions, it was reported that the use of the da/dN-ΔK_eff_ relationship is effective for the analysis of FCG behavior before and after R fluctuation. When a large-scale yielding phenomenon is included, as in the present study, it is effective to use the da/dN-ΔJ_eff_ relationship for the analysis of FCG behavior before and after the R fluctuation.

## 5. Conclusions

Using the extruded texture of the AZ31Mg alloy under the constant R condition of program [I] and the R fluctuation experiment of program [II] were conducted, and the FCG behavior and mechanism at that time were studied. The results are summarized as follows:(1)The FCG velocity accelerated due to the fatigue load on the compression side at R = −1 after R fluctuation. The degree of acceleration was greater in condition (B) than in condition (A), and the higher the K_max_ value, the faster the acceleration.(2)The hysteresis area of the cyclic σ-ε curve after R fluctuation was wider in condition B than in condition A and was wider when the K_max_ value was larger. In addition, the hysteresis area on the compression side was larger than that on the tension side, indicating asymmetry. The magnitude of the hysteresis loop area corresponded to the degree of acceleration of the FCG speed owing to the R fluctuation.(3)From the observation of fatigue cracks on the specimen surface after R fluctuation, twins were formed around the cracks owing to the compressive load at R = −1, and large deformation occurred. In addition, numerous microcracks are generated at the twins.(4)The following two acceleration mechanisms can be mentioned as the source of the FCG velocity at R = −1 after R fluctuation. First, the crack opening displacement increases due to the twinning caused by the compressive load of R = −1, and the FCG driving force increases. The second is the coalescence of the main crack and a plurality of microcracks generated on the twins.(5)The effective ΔJ integral range, ΔJ_eff_, was approximately evaluated using the cyclic σ-ε curve of the fatigue process. Based on the consistency with other experimental results, the evaluation results are considered valid.(6)Under condition (B), a large number of tensile twins were generated under the compression load after the R fluctuation, and as a result, the crack tip neighborhood changed to a large-scale yielding state. The FCG velocity after R fluctuation was controlled by ΔJ_eff_.

## Figures and Tables

**Figure 1 materials-15-00755-f001:**
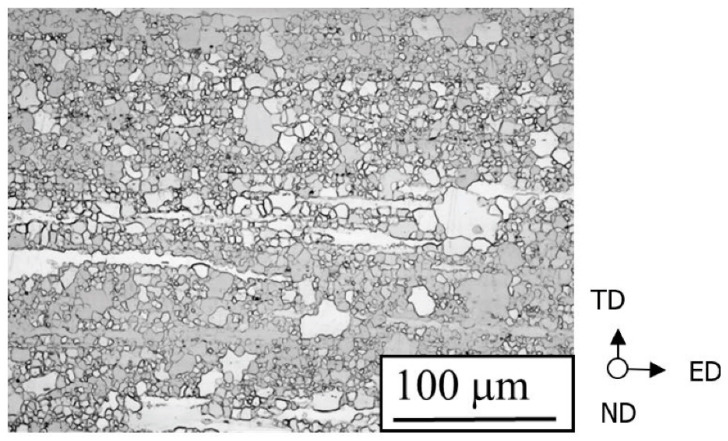
Microstructure of the Mg alloy AZ31 used in the present study.

**Figure 2 materials-15-00755-f002:**
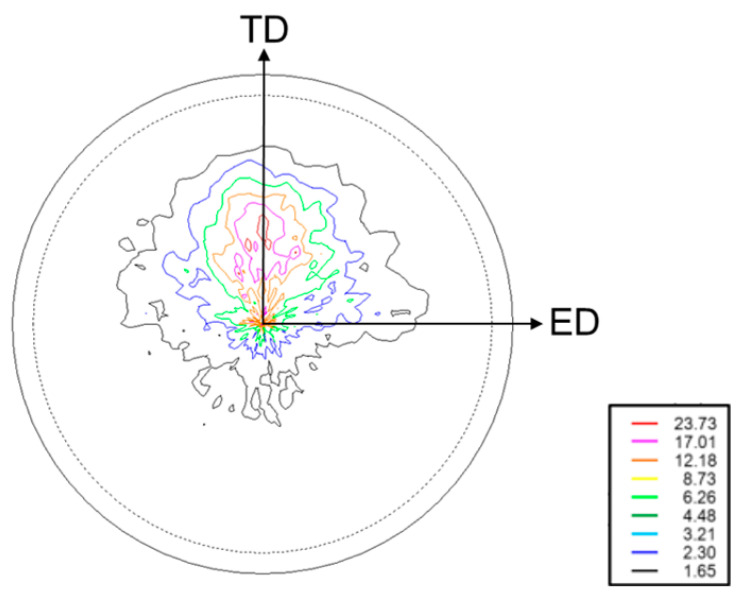
Pole figure for the basal plane {0002}.

**Figure 3 materials-15-00755-f003:**
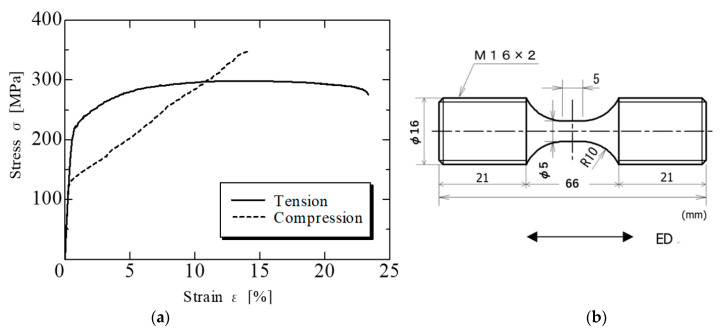
Monotonic tension and compression tests. (**a**) Monotonic stress–strain curve test. (**b**) Specimen used for the monotonic test.

**Figure 4 materials-15-00755-f004:**
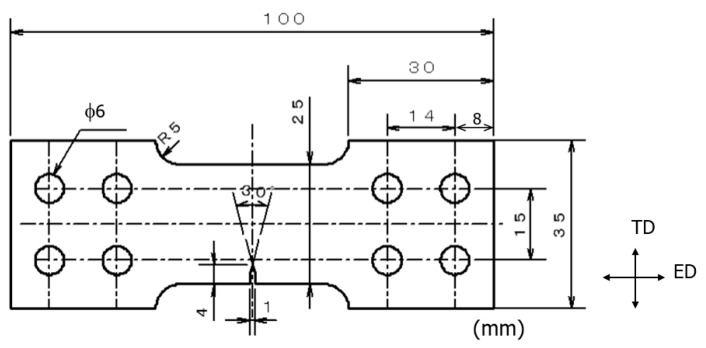
Shape and dimensions of the SENT specimens (mm) for FCG observation of long through-thickness cracks.

**Figure 5 materials-15-00755-f005:**
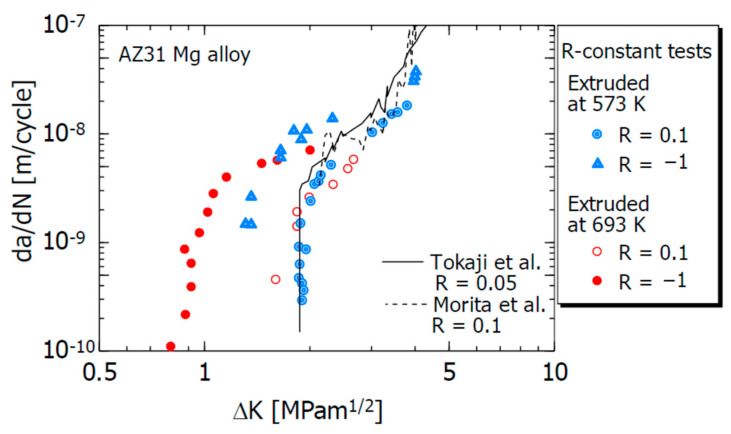
Relations between da/dN and ΔK.

**Figure 6 materials-15-00755-f006:**
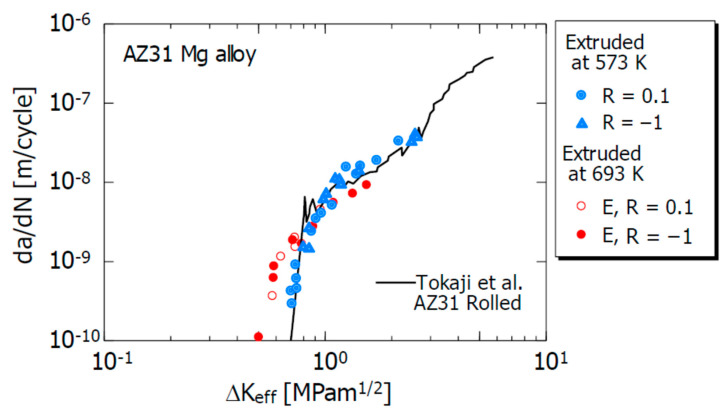
Relations between the rate of crack propagation da/dN and effective stress intensity factor range ΔK_eff_.

**Figure 7 materials-15-00755-f007:**
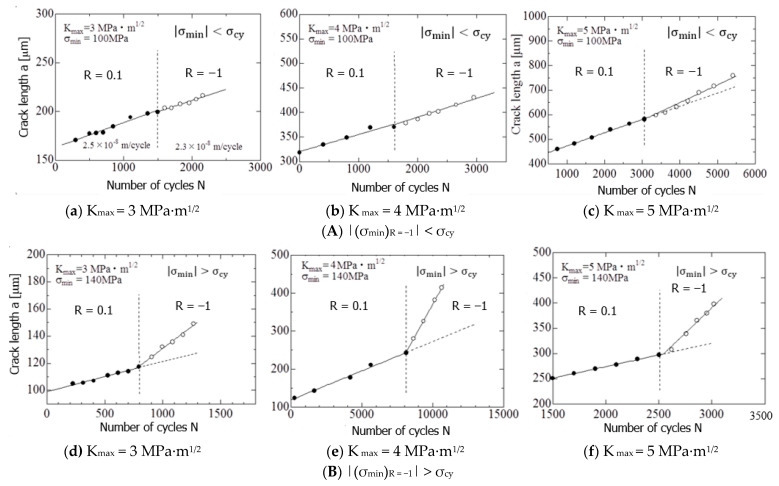
FCG behavior before and after R ratio change under the conditions (**A**) and (**B**). (SENT specimen was used.).

**Figure 8 materials-15-00755-f008:**
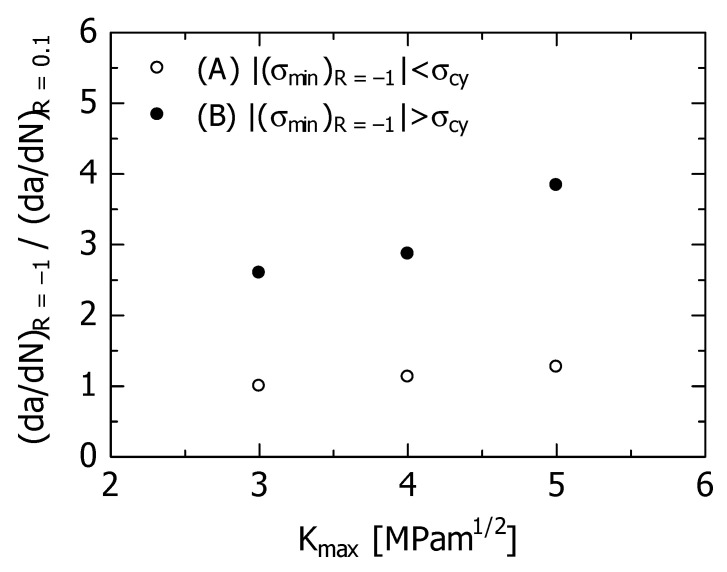
Effect of K_max_ on (da/dN)_R = −1_/(da/dN)_R = 0.1_ due to R fluctuation.

**Figure 9 materials-15-00755-f009:**
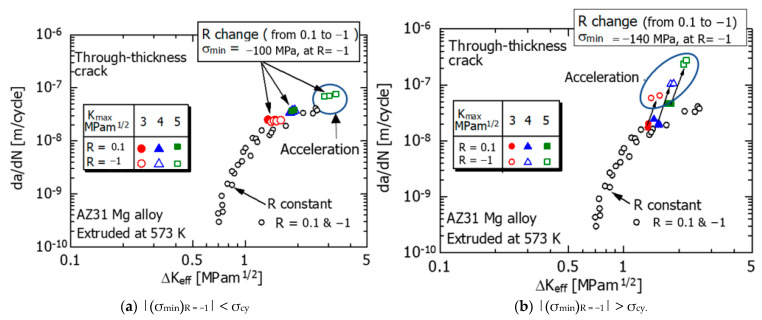
da/dN–△K_eff_ relationship in the R fluctuation experiment. (**a**) |(σ_min_)_R = −1_| < σ_cy_ (**b**) |(σ_min_)_R = −1_| > σ_cy_.

**Figure 10 materials-15-00755-f010:**
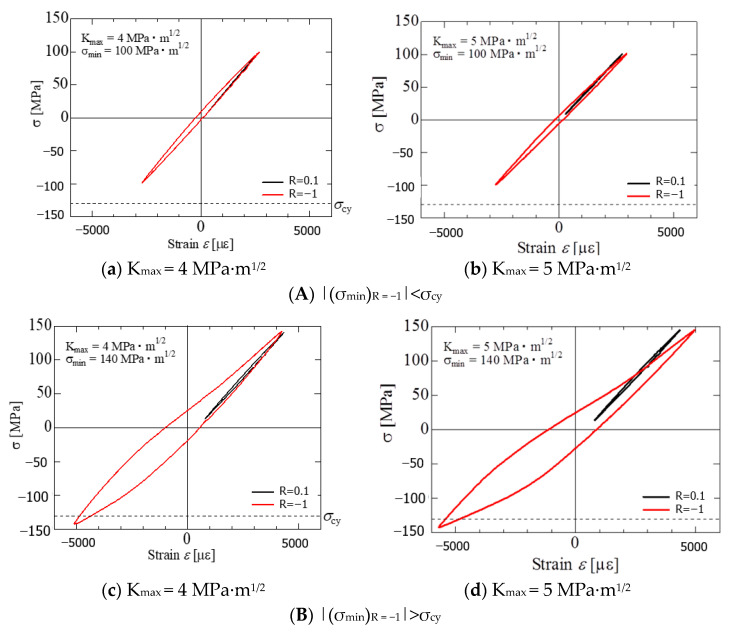
Hysteresis loop due to R-ratio change (SENT specimen). (**A**) |(σ_min_)_R = −1_|<σ_cy._ (**B**) |(σ_min_)_R = −1_|>σ_cy_.

**Figure 11 materials-15-00755-f011:**
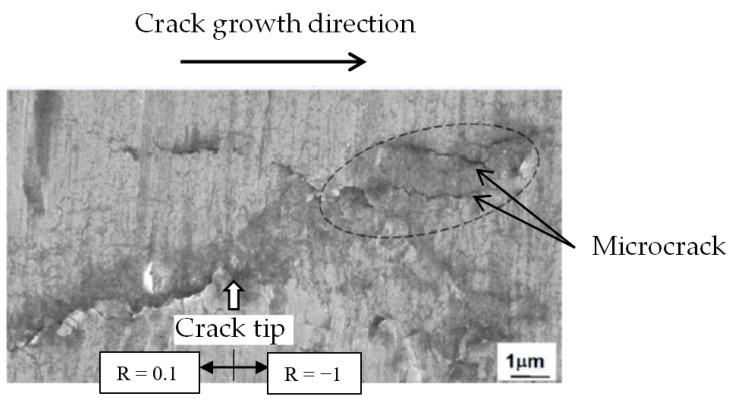
Change in the through-thickness crack morphologies by the R-ratio change (from 0.1 to −1) at the constant K_max_ values (5 MPa∙m^1/2^, Condition B |(σ_min_)_R = −1_| > σ_cy_).

**Figure 12 materials-15-00755-f012:**
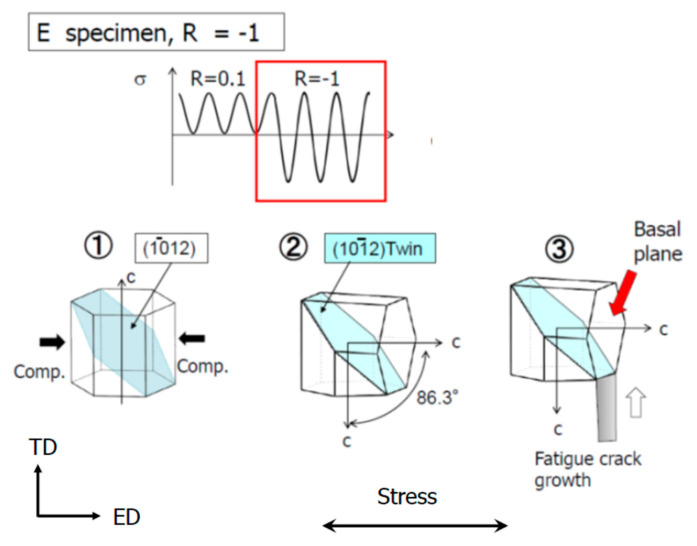
FCG velocity acceleration mechanism by compressive load of R = −1.

**Figure 13 materials-15-00755-f013:**
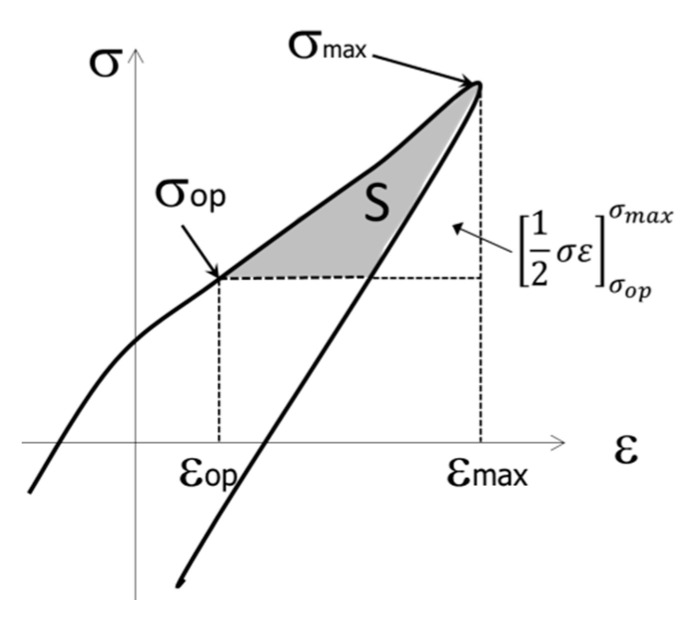
Definition of S in the hysteresis loop, σ_op_ ≤ σ ≤ σ_max_.

**Figure 14 materials-15-00755-f014:**
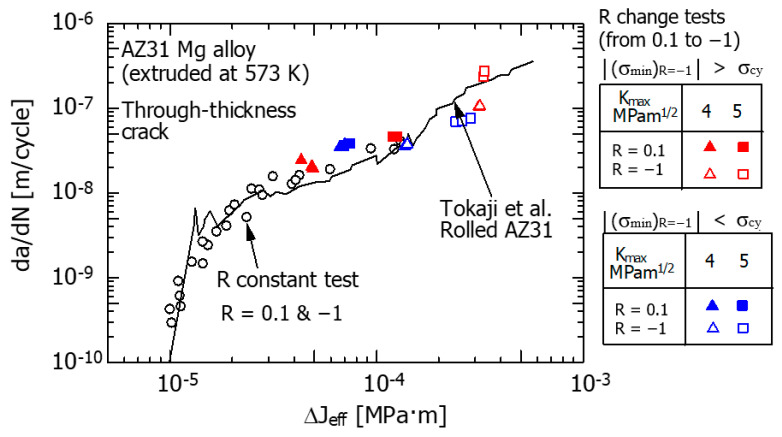
Relations between da/dN and ΔJ_eff_ (SENT).

**Table 1 materials-15-00755-t001:** Chemical compositions of the Mg alloy AZ31 used. [wt%].

Al	Zn	Mn	Fe	Si	Pb	Cu	Ni	Mg
3.04	0.85	0.33	0.0031	0.014	0.0033	0.0017	0.0006	Bal.

**Table 2 materials-15-00755-t002:** Mechanical properties of the specimen used.

Specimen Orientation	Loading Direction	Yield Strength MPa	Breaking Strength MPa	Young’s Modulus, E	Elongation %
ED	Tension	σ_ty_ = 210	298	45 GPa	21.8
	Compression	σ_cy_ = 130	-	45 GPa	-

**Table 3 materials-15-00755-t003:** Acceleration rate of FCG speed when R fluctuates from 0.1 to −1.

(**A**) |(σ_min_)_R = −1_| < σ_cy_			
**K_max_ [MPa∙m^1/2^]**	**(da/dN)_R = 0.1_ [m/cycle]**	**(da/dN)_R = −1_ [m/cycle]**	**Acceleration Rate (da/dN)_R = −1_/(da/dN)_R = 0.1_**
3	2. 5 × 10^−8^	2.5 × 10^−8^	1. 0
4	3.4 × 10^−8^	3.8 × 10^−8^	1.13
5	5.5 × 10^−8^	7.0 × 10^−8^	1.27
(**B**) |(σ _min_)_R = −1_| > σ_cy_			
**K_max_ [MPa∙m^1/2^]**	**(da/dN)_R = 0.1_ [m/cycle]**	**(da/dN)_R = −1_ [m/cycle]**	**Acceleration Rate (da/dN)_R= −1_/(da/dN)_R= 0.1_**
3	2.5 × 10^−8^	6.5 × 10^−8^	2.6
4	3.0 × 10^−8^	8.6 × 10^−8^	2.87
5	5.0 × 10^−8^	19.2 × 10^−8^	3.84

## Data Availability

The data presented in this study are available on request from the corresponding author.

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
