# Peer review of "Elasto-Plastic Fatigue Crack Growth Behavior of Extruded Mg Alloy with Deformation Anisotropy Due to Stress Ratio Fluctuation"

_materials, 2022, doi:10.3390/ma15030755_

Round 1

Reviewer 1 Report

The authors conduct a series of interesting experiments to ascertain the rate of fatigue crack growth under a variety of conditions.  The results are impressive and exciting because they are clearly in line with those of previous researchers, confirming the accuracy and significance of their findings. 

Regarding minor points regarding the experimental section, this reviewer would be surprised in the authors really used emery paper to smooth the surface of the test pieces.  In Western metallurgy emery paper has not been seen for over 50 years. It seems likely that the abrasive paper was in fact SiC paper.

The interesting results shown in Figure 7 were practically impossible for a reader to understand clearly; the scales were so different that the graphs could not easily be compared.  This reviewer sketched out a combined figure for results Fig 7 d,e,f, using crack length scale 0-500 um and number of cycles scale from 0 - 15000.  However, the different starting lengths obscured the hoped-for simplicity of comparing the curves.  If some common starting point were possible (by perhaps plotting 'reduced' values of crack length, as dimensionless numbers) a clear comparison would be valuable and appreciated by most readers. Figs 7 abc might also benefit from a presentation to clarify a comparison between the curves. 

Moving on to more serious issues, the authors appear unaware of the new approach to crack growth mechanisms which is becoming discussed and gradually accepted in Europe and the USA. 

The new approach draws attention to the creation of cracks in the liquid metal because of the folding in of the surface oxide on the liquid during the casting of the original ingot material. The folding ensures dry oxide surface to dry oxide surface, so the lack of bonding between the two films creates a crack.  The cracks, generally called 'bifilms', survive plastic working in the solid state, but tend, of course, to align with the working direction. Grain growth during working and heat treatment is arrested at bifilms (grains cannot cross the 'air gap' of the bifilm central interface) so that grain growth is chaotically influenced, causing the huge variability clearly seen in the authors' Fig 1. (How can this great variability be explained otherwise?)

The dense population of bifilms in the alloy  would be expected to create the hysteresis loops when R = -1 in Fig 9. In Fig 10 the so-called 'microcracks' are almost certainly the pre-existing bifilm cracks simply opening under tensile strain. The cracks are 5 - 10 um long, which is approximately the same as the grain size in Fig 1. This is a typical bifilm size for this type of small ingot. The association of cracks and twins is not that the twin create stress and so create cracks, but is the result of twins automatically forming on a bifilm, because the central air gap of the pre-existing bifilm crack permits the reduction of the strain energy of formation (the change of volume and shape are accommodated by deformation into the pre-existing space in the matrix, avoiding the high energy required for the plastic deformation of the matrix.)

There is now a substantial literature in the metallurgical and casting journals, and several substantial casting handbooks on this new approach.  The authors are recommended to become familiar with this new interpretation of crack growth. It appears to be powerful in the clarity of its explanations. Furthermore, it seems that crack growth might be preventable by improved casting technology!  These are revolutionary possibilities!

The above suggestions are not mandatory, but the authors may find the new suggested mechanisms apply accurately to their work, and would bring their paper into line with the latest thinking in this field. 

Author Response

To Professor:

We would like to express our sincere gratitude to you for scrutinizing this paper and providing valuable opinions regarding its improvement. We have improved the paper according to your comments. We would appreciate it if you allowed for our paper to be published in Materials.

                        Dr. Kenichi Masuda

Reviewer 2 Report

The submitted manuscript entitled ‘Elasto-plastic fatigue crack growth behavior of extruded Mg alloy with deformation anisotropy due to stress ratio fluctuation’ is dealing with the experimental fatigue crack propagation measurement of an extruded Mg alloy, containing Al, Zn and Mnas main alloying elements. The manuscript is interesting and worth publishing. However, several issues (including three significan concerns) listed below should be carefully addressed.

  1. Please give official (not commercial, like Gmail) e-mail address for all authors.
  2. Introduction: most of the cited references are mentioned only, but their main findings are missing.
  3. The compressive strength cannot be negative (only compressive stress can be); therefore, the absolute value sign can be omitted regarding the ‘conditions’.
  4. How was the chemical composition in table 1 determined?
  5. What is the difference between the directions [0002] and [0001], are not they both belongs to the <0001> directions?
  6. How can be TD and ND directions determined for a round bar?
  7. Fig 1: The microstructure is quite uneven. This may have significant effect on the fatigue life. Please discuss in detail. This is the first main concerns of this Reviewer.
  8. If axes c is perpendicular to the ED, the basal plane should be parallel to the ED. In fig 2 this does not hold. Please discuss.
  9. Compressive test should be done on cylindrical samples between flat plates, not on dogbone like samples.
  10. How was the strain measured?
  11. What kind of testing machine was used to perform the fatigue tests?
  12. Fig 3a: how many measurements are behind the curves? How were the stress and strain calculated (engineering or true system)?
  13. Fig 3b: the drawing is technically incorrect. Diameter sign is missing. Dimensions are missing.
  14. Table 2: please add standard deviations.
  15. Fig 4: the drawing is technically incorrect. Diameter sign is missing. Dimensions are missing. There are Japaneese characters in the figure.
  16. ΔK is depending on a. If a is measured by replica method, not continuously, how were the proper ΔK settings (increasing, decreasing and constant) set? This is the second main concern of this Reviewer.
  17. Please add ‘The simple evaluation formula proposed by Rice et al. [32]’ as an equation.
  18. As turned out from section 3.1, two different extrusion temperatures were considered (573 and 693 K). This may have effect on the microstructure and the mechanical properties of the material as well. However, they are not reported in section 2, previously.
  19. Please do not use ‘×’ to indicate multiplication, since it is reserved for cross-product. Please use ‘∙’ instead.
  20. Fig 8a contains a typo (‘Acceleratio’).
  21. Fig 5: acceleration for 693 K extruded material is not obvious on R. The red curves are at the same height. Please explain.
  22. Line 284: Fig. 7 (B), correctly.
  23. Please use ‘dash line’ or ‘dashed line’ instead of ‘broken line’.
  24. Figure 9: loops are changing with the number of cycles, please present this alternation and plot more loops at different number of cycles.
  25. ‘…a fatigue load acts approximately perpendicular to the basal plane {0002} so that the basal plane dislocation operates.’ – please explain, normal stresses are not indicating dislocation slipping.
  26. Eq 2: please rearrange for clarity. Is (W-a) in the nominator or denominator?
  27. Is Eq 2., developed for plates with center crack holds for SENT samples (do not think so)? Please clarify and explain. This is the third main concern of this Reviewer.
  28. Line 509: Fig. 12, correctly.
  29. It is unclear in which cycle was S calculated.

Author Response

(The authors gave the same response as above.)

Round 2

Reviewer 2 Report

Thank you for all the changes and corrrections.